# Investigating the Role of BATF3 in Grass Carp (*Ctenopharyngodon idella*) Immune Modulation: A Fundamental Functional Analysis

**DOI:** 10.3390/ijms20071687

**Published:** 2019-04-04

**Authors:** Denghui Zhu, Rong Huang, Peipei Fu, Liangming Chen, Lifei Luo, Pengfei Chu, Libo He, Yongming Li, Lanjie Liao, Zuoyan Zhu, Yaping Wang

**Affiliations:** 1State Key Laboratory of Freshwater Ecology and Biotechnology, Institute of Hydrobiology, Chinese Academy of Sciences, Wuhan 430072, China; denghuizhu123@163.com (D.Z.); huangrong@ihb.ac.cn (R.H.); ppf@ihb.ac.cn (P.F.); liangmingchen@126.com (L.C.); luolifei145@163.com (L.L.); chupengfei17@163.com (P.C.); helibowudi@ihb.ac.cn (L.H.); liym@ihb.ac.cn (Y.L.); liaolj@ihb.ac.cn (L.L.); zyzhu@ihb.ac.cn (Z.Z.); 2University of Chinese Academy of Sciences, Beijing 100049, China; 3Innovative Academy of Seed Design, Chinese Academy of Sciences, Beijing 100101, China

**Keywords:** Grass carp, BATF3, Innate immune, Subcellular localization, Gene silencing

## Abstract

Basic leucine zipper transcription factor ATF-like (BATF)-3, belonging to activator protein 1 (AP-1) superfamily transcription factors, is essential for homeostatic development of CD8α^+^ classical dendritic cells activating CD8 T-cell responses to intracellular pathogens. In this study, the characteristics and cDNA cloning of the *CiBATF3* molecule were described in grass carp (*Ctenopharyngodon idella*). *CiBATF3* had abundant expression in immune-related organizations, including liver, spleen and gill, and grass carp reovirus (GCRV) infection had significantly changed its expression level. After *Ctenopharyngodon idella* kidney (CIK) cells were challenged with pathogen-associated molecular patterns (PAMPs), polyinosinic:polycytidylic acid (poly(I:C)) stimulation induced higher mRNA levels of *CiBATF3* than that of lipopolysaccharide (LPS). Subcellular localization showed that CiBATF3-GFP was entirely distributed throughout cells and nuclear translocation of CiBATF3 was found after poly(I:C) treatment. Additionally, the interaction between CiBATF3 and interleukin 10 (IL-10) was proven by bimolecular fluorescence complementation (BiFC) system. The small interfering RNA (siRNA)-mediated *CiBATF3* silencing showed that the mRNA of *CiBATF3* and its downstream genes were down-regulated in vitro and in vivo. *CiBATF3* played a negative regulatory role in the transcriptional activities of AP-1 and NF-κB reporter gene. In summary, the results may provide valuable information on fundamental functional mechanisms of *CiBATF3*.

## 1. Introduction

The basic leucine zipper transcription factor ATF-like (BATF) family consists of BATF, BATF2 and BATF3 [1], belonging to the activator protein 1 (AP-1) family of transcription factors [2]. An alpha-helical bZIP domain containing a leucine zipper motif and a basic DNA binding region is characteristic of members of this family [3,4,5]. In every tissue and cell of the immune system, bZIP proteins are crucial transcriptional regulators by interactions with AP-1-binding elements. Two of the earliest identified bZIP proteins, FBJ osteosarcoma oncogene (FOS) and JUN (AP-1 transcription factor subunit), constituting the heterodimeric transcription factor termed AP-1 [6], can widely regulate gene expression. It is known that the leucine zipper motif is essential for interaction with a bZIP protein or a non-bZIP transcription factor such as interferon regulatory factors (IRF) that modulates target genes [7].

Dendritic cells (DCs) are hematopoietic cells that belong to the antigen-presenting cell (APC) family. Their primary function is to process antigens and present them to T cells to promote immunity to foreign antigens and tolerance to self-antigens [8]. Mouse DCs are subdivided into only three main subtypes: two main lineages of classical DCs (cDCs) (“classical type 1 DCs (cDC1s)” for CD8α^+^ and CD103^+^ DCs, and “cDC2s” for CD11b^+^ and CD172a^+^ DCs based on their distinct developmental pathways) and plasmacytoid DCs (pDCs) [9]. The lineages of cDC1s include both lymph node (LN)-resident and migratory subsets [10,11]. Lymphoid tissue-resident cDCs differentiate in, and spend their entire lives within, lymphoid tissues [12]. Tissue-migratory cDCs are located in the peripheral lymph nodes (LNs). They refer to nonlymphoid tissue DCs that have migrated to the tissue-draining LNs through the lymphatics as opposed to the blood-borne lymphoid-resident DCs [13]. The resident and migratory cDC1s cells are either CD8α^+^ (LN resident) or CD103^+^ (migratory). cDC1 development is mediated by the transcription factors IRF8 and BATF3. Mice lacking IRF8 [14] or BATF3 [15] exhibit a severe defect in the development of cDC1s. In particular, IRF8 mutant and IRF8-deficient mice lack CD8α^+^ cDCs but have additional immune defects [16,17,18]. In BATF3^−/−^ mice, DCs lack virus specific CD8^+^ T cell in responses to West Nile virus and also have a defect in cross-presentation [15]. Migratory DCs mediate the exchange of archived antigens between lymphatic endothelial cells (LECs) and antigen-presenting cells (APCs). After vaccination, antigen exchange and cross-presentation are performed by both BATF3-dependent and BATF3-independent DCs [19]. In adult and neonatal mice, the intestinal BATF3-dependent DC is vital for optimal antiviral T-cell responses [20]. Recently, many studies [15,21,22] have shown that BATF3 plays very important roles in promoting expansion of functional CD8^+^ cDCs to control infection of intracellular pathogens, and via the interaction of the conserved LZ domain with IRF4 or IRF8, this function may be compensated by other members of the BATF family [22].

Up to now, research on BATF3 has mostly focused on mammals rather than fish. In fish, the BATF3 genes have been identified in *Danio rerio* [23], *Salmo salar* [24], and *Oncorhynchus mykiss* [25]. In China, Grass carp is one of the most important economic freshwater aquaculture species, and deeply studying its functional genes [26,27,28], whole-genome [29], transcriptome [30,31,32] and microRNA regulation [33] has provided new data for studying the immune system of fish. In our study, we amplified the full-length cDNA sequence of *Ctenopharyngodon idella* BATF3 (*CiBATF3*) and, to a better understanding of the role of *CiBATF3* gene in fish immune responses, sequence analysis and functional experiments of *CiBATF3* were also completed. The bioinformatics and functions analysis of *CiBATF3* will help for studying the transcriptional regulation by BATF3 in fish immune systems in the future.

## 2. Results

### 2.1. Analysis of CiBATF3

The full-length cDNA of *CiBATF3* is 1198 bp with a 5′-terminal untranslated region (UTR) of 147 bp, a 3′-UTR of 688 bp, two mRNA instability motifs (ATTTA), a polyadenylation signal (AATAA), and an open reading frame of 363 bp encoding for 120 amino acid residues with a predicted molecular mass of 13.884 kDa and theoretical isoelectric point of 8.54. SMART analysis revealed that a typical basic leucine zipper (bZIP, 30–87 aa) domain was discovered in the deduced amino acid of *CiBATF3* (Appendix A). No signal peptide was detected by the signalP program. The genomic sequence of *CiBATF3* ORF, which contained three exons and two introns, was 1621 bp (Figure 1A) and all the intron–exon boundaries displayed the conserved GT/AG rule. The bZIP domain was composed of a leucine zipper (LZ), a hinge (H) region and a DNA binding domain (DB), which existed in all the BATF3. The amino acid sequences of these domains are highly homologous with six absolutely conserved leucine residues. The *N*- and *C*-terminal regions of *CiBATF3* have a relatively low sequence homology (Figure 1B). Swiss-model analysis indicated the *CiBATF3* protein contained a single α-helix, similar to that of human and zebrafish (Figure 1C). A comparison of homology revealed that the deduced *CiBATF3* shares 36.9–85% sequence similarity with BATF3 of other vertebrate, with it being the most similar to that of the *Danio rerio* (85% sequence similarity), followed by the *Ictalurus punctatus* (69.8% sequence similarity) (Table 1).

To detect the evolutionary conservation in genomic structure, the syntenic analysis of *CiBATF3* genes was performed with neighboring genes from other species. The result show that the upstream genes of higher vertebrates BATF3 had relatively conserved synteny, whereas *CiBATF3* had highly conserved synteny with other fish BATF3 genes (Figure 2A). Especially, the syntenic relationship of the BATF3 and ATF3 genes in the genome is highly conserved among the selected species.

To determine the molecular evolutionary relationship, BATF family amino acid sequences from various vertebrate species were used to construct a phylogenetic tree. As shown in Figure 2B, three major clades corresponding to BATF1s, BATF2s and BATF3s were detected. All the fish BATF3 proteins clustered together, of which the *CiBATF3* branched most closely to *Danio rerio* BATF3.

### 2.2. The mRNA Expression Levels of CiBATF3 in Tested Tissues

The spatial mRNA expression profiles of *CiBATF3* in all tested tissues (skin (SK), middle kidney (MK), intestine (I), liver (L), gill (G), heart (H), spleen (S), head kidney (HK), muscle (M) and brain (B)) are shown in Figure 3A. The *CiBATF3* had the dominant abundance expression in liver, followed by spleen, gills, heart and head kidney, and was low in skin.

### 2.3. Analysis of CiBATF3, IRF8 and IL-10 Genes After GCRV Infection In Vivo

The temporal expression of the *CiBATF3* gene in six immune tissues after GCRV challenge is shown in Figure 3B–G. In the spleen, intestine and head kidney, *CiBATF3* were down-regulated on Day 1, and then were up-regulated on Days 3 and 5. In the middle kidney, the *CiBATF3* expression levels were promoted during the entire detection period. Expression of GCRV S6 gene increased in gill and intestine on Days 3 and 5 after injection (Figure 3H), which verified infection had occurred.

To find the antiviral mechanism in grass carp after GCRV infection, the expression levels of IRF8 (Figure 4A–D) and IL-10 (Figure 4E–H), which were regulated by *CiBATF3*, were detected in four immune tissues (spleen, intestine, head kidney and middle kidney). After GCRV infection, interestingly, the response patterns of IRF8 and IL-10 were up-regulated in the four examined tissues similar to *CiBATF3*, respectively.

### 2.4. Time-Course Analysis of CiBATF3 Expression in CIK Cells After LPS Exposure or Poly(I:C) Challenge

Time-course expression of *CiBATF3* expression in CIK cells after LPS exposure is shown in Figure 5A. The expression level of *CiBATF3* gradually increased from 0 h to 6 h, reached the peak at 6 h (1.54-fold, *p* < 0.05), and then decreased until 48 h. Upon poly(I:C) stimulation, the *CiBATF3* transcripts were induced from 3 h to 48 h (Figure 5B), and reached peak at 36 h, at which time the expression level was 18.36-fold (*p* < 0.01) compared to the control group (0 h).

### 2.5. siRNA-Mediated CiBATF3 Silencing Could Affect the Expression of its Downstream Molecules In Vitro and In Vivo

The effect of siRNA-mediated *CiBATF3* silencing on the expression of its downstream genes was examined by RT-qPCR analysis, including GATA binding protein 3 (GATA3), interleukin 4 (IL-4), interleukin 10 (IL-10), interferon regulatory factor 8 (IRF8), interleukin 12a (IL-12 p35), interleukin 12b (IL-12 p40) and cellular myelocytomatosis oncogene (*c-myc)*, because it has been reported that these genes are regulated by BATF family members and involved in immune response [7,22,35,36]. In the in vitro assay, siRNA transfection depressed BATF3 mRNA level in CIK cells (Figure 6A). In this condition, the mRNA levels of IL-4, IL-10 and IL-12 p40 displayed a 0.49-fold, 0.13-fold and 0.15-fold decrease, respectively (Figure 6B). However, these genes, including GATA3, IRF8, IL-12 p35 and *c-myc*, did not change significantly compared to negative the control group.

For in vivo test, the *CiBATF3* mRNA were down-regulated 0.17-fold (*p* < 0.01) and 0.33-fold (*p* < 0.01) in middle kidney (MK) and head kidney (HK), respectively (Figure 7A), after siRNA silencing of *CiBATF3*. Most of its downstream genes were significantly down-regulated in middle kidney and head kidney except that IRF8 and IL-12 p35 did not change significantly in head kidney at this condition (Figure 7B–H).

### 2.6. Cell Transfection and Luciferase Activity Analysis

To investigate the function of CiBATF3, HEK293 cells were co-transfected with empty vector or pCDNA3.1-CiBATF3 together with NF-κB or AP-1 reporter gene. As shown in Figure 8, these results indicated that CiBATF3 can suppress the transcriptional activities of AP-1 and NF-κB reporter gene.

### 2.7. Subcellular Localization of CiBATF3 in CIK Cells and HEK293 Cells

The subcellular localization of the CiBATF3 protein was observed using confocal microscope in CIK cells or HEK293 cells. In CIK cells, CiBATF3-pEGFP was strongly dispersed throughout the cytoplasm and the nuclear areas, which was consistent with pEGFP-N3 (Figure 9A). In HEK293 cells, CiBATF3-pEGFP was also distributed same as that of CIK cells (Figure 9B).

### 2.8. The CiBATF3-GFP Nuclear Translocation Induced by Poly(I:C) Stimulation

As shown in Figure 9C, PAMPs induce the relocation of CiBATF3 to varying degrees. In poly(I:C) stimulated-cells, CiBATF3 proteins appeared to have a tendency to concentrate in the nucleus. After LPS stimulation, the protein expression pattern of CiBATF3 did not change.

### 2.9. Interaction between CiBATF3 and IL-10

The interaction of CiBATF3 and IL-10 in living cells was visualized by the mNeptune-based BiFC system. No fluorescence signal was observed when pBATF3-MN155 or pMC156-IL-10 was transfected into CIK cells alone. However, after co-transfection of pBATF3-MN155 and pMC156-IL-10, the red mNeptune fluorescence signal in the cytoplasm was bright red (Figure 9D). Thus, it confirmed the interaction between CiBATF3 and IL-10 in CIK cells.

## 3. Discussion

It is known that BATF3 is crucial for regulating numerous cellular processes [7] and immune-specific functions [15,21,37,38]. In fish, BATF3 signaling attracts attention [25]. Here, a BATF family member (*CiBATF3*) was firstly cloned and identified in *C. idella*. Multiple sequence alignments, domain structure and phylogenetic analyses strongly showed functional homology of *CiBATF3* orthologs in other species. The high expressions of *CiBATF3* in immune-related tissues (liver, spleen, gills and head kidney) suggested that it plays critically important roles in the immune system of grass carp. Thus, expression pattern of *CiBATF3* gene post GCRV, LPS and poly(I:C) stimulation were further analyzed.

Toll-like receptors (TLRs) [39,40,41,42,43,44] play a key role in the innate immune response in mammals by identifying conserved molecular patterns associated with different microorganisms. TLR4 has been identified as a signaling molecule essential for the responses to LPS [45]. TLR3 can be stimulated by double-stranded RNA [46]. Poly(I:C) is well known as a dsRNA analog. In rainbow trout, Poly(I:C) and R848 stimulation dramatically up-regulate BATF3a and BATF3b. After LPS treatment, BATF3a is also up-regulated to some extent. However, no significant changes in BATF3b are detected in cells treated with LPS and Flagellin [25]. This indicates that BATF3 may be involved in the TLRs signaling pathway. In this study, the mRNA levels of *CiBATF3* were induced by poly(I:C) and LPS. However, poly(I:C) can induce higher mRNA levels of *CiBATF3* than LPS. Moreover, *CiBATF3* proteins localization appeared to have a tendency to concentrate in the nucleus following poly(I:C) stimulation but there was no change after LPS stimulation (Figure 9C). Therefore, these suggest that *CiBATF3* may be more sensitive to the TLR3 signaling pathway than TLR4 signaling pathway.

It is reported that BATF family members have the function of regulating the gene expression. In BATF^−/−^ BATF3^−/−^ T_H_2 cells, the expressions of IL-4 and IL-10 are significantly lost, indicating that BATF and BATF3 act together on the expression of these two cytokine genes [22]. BATF contributes to express the major T_H_2 cell-associated transcription factor GATA3 and also has direct effects on the genes encoding other T_H_2 cell effector factors (e.g., cytotoxic T lymphocyte antigen 4 (CTLA4), IL-4 and IL-10). BATF3 also helps regulate these T_H_2 cell-associated genes [7]. STAT-mediated BATF3 expression is crucial for lymphoma cell survival and promotes MYC activity in classical Hodgkin lymphoma and anaplastic large cell lymphoma, and a new oncogenic axis is recognized in these lymphomas [35]. The evidence for the direct role of BATF3 in regulating IRF8 expression has been discovered. In CD24^+^ cDCs, the auto-activation of IRF8, dependent on BATF3 may operate at an element +32 kb from the IRF8 transcriptional start site (TSS), within an IRF8 superenhancer [36]. In this study, siRNA-mediated *CiBATF3* silencing showed that *CiBATF3* can regulate most of the tested gene expression levels.

According to many studies, transcriptional regulator NF-κB can activate downstream anti-apoptotic genes and promote cell survival [47,48,49,50]. AP-1 has been proven to promote cell proliferation and malignant transformation [27,51,52]. In mice, BATF negatively regulates the transactivation of an AP-1 reporter gene in vivo [53]. BATF2 inhibites both AP-1 and NF-κB reporter gene transcriptional activities [54]. In our study, CiBATF3 negatively regulated the transcriptional activities of AP-1 and NF-κB reporter gene. Therefore, we speculate that CiBATF3 protein can inhibit cell proliferation by inhibiting AP-1 activity. Inhibition of NF-κB activity by CiBATF3 is likely to attenuate the effects of anti-apoptosis, thereby enhancing the effects of pro-apoptotic.

In conclusion, the results of this study may provide valuable information for the study of the potential functional mechanism of BATF3 in teleosts.

## 4. Materials and methods

### 4.1. Cells, Plasmid and Fish

The HEK293 cells were cultured in high glucose Dulbecco’s modified Eagle’s medium (DMEM; Hyclone, Logan, UT, USA), with 10% FBS, 100 mg/mL streptomycin (Sigma) and 100 IU/mL penicillin (Sigma) under a humidified condition with 5% CO_2_ at 37 °C. The CIK cells (provided by China Center for Type Culture Collection, Wuhan, China) were cultured under a humidified condition with 5% CO_2_ at 28 °C, using Medium 199 (Sigma, St. Louis, MO, USA) with 10% FBS, 100 mg/mL streptomycin (Sigma) and 100 IU/mL penicillin (Sigma). The pEGFP-N3 plasmid was previously in our lab. The pMN155 and pMC156 plasmids were given by Professor Zongqiang Cui, Wuhan Institute of Virology, Chinese Academy of Sciences. pMD18-T were obtained from TransGen Biotech (Beijing, China). pRL-TK, pGL3-bacic, pCDNA3.1 (-), pNFқB-TA-luc, and pAP1-TA-luc were purchased from Beyotime (Shanghai, China). Grass carps (3 months old; weight, 10 ± 2 g; length, 7 ± 3 cm) were collected at the GuanQiao Experimental Station, Institute of Hydrobiology, Chinese Academic of Sciences, and acclimatized in aerated freshwater at 28 °C for one week. The fish were fed with a commercial feed (Tong Wei, Chengdu, China) to adapt to the environment until 24 h before the experiments under the same conditions.

### 4.2. GCRV Infection and Sampling

For the tissue distribution experiment, tissues including liver, skin, intestine, gill, spleen, muscle, heart, head kidney, brain and middle kidney were collected from five healthy grass carp.

The GCRV infection experiment was carried out as described previously [55] with modifications. Five healthy grass carps were injected intraperitoneally with 200 μL PBS (pH 7.4) as control group while each fish of challenge group was injected with an equal volume of GCRV (GD108 strain). The titer of virus detected by RT-qPCR (the special primers are listed in Table 2) was 3.12 × 10^3^ copy/μL. These injected fish were kept under the same conditions as mentioned above with commercial feed during the test. Five individuals were collected at 0, 1, 2, 3, 4, 5 and 6 days post injection (dpi). The head kidney, gill, liver, intestine, middle kidney and spleen were harvested in TRIzol reagent (Invitrogen, Carlsbad, CA, USA) and stored at −80 °C until RNA extraction.

Total RNAs were extracted and the first-strand cDNA was synthesized using ReverTra Ace kit (Toyobo, Osaka, Japan). The reaction was performed at 42 °C for 1 h, and terminated by heating at 95 °C for 5 min.

### 4.3. Cloning the Full-Length CiBATF3 cDNA

The specific fragments with the coverage of open reading frame (ORF) region were obtained by blasting the *CiBATF3* sequences of zebrafish (Accession no. NM_001045392.1) with the *C. idella* transcriptional database [29]. Then, the ORF sequence was amplified and then the 5ʹ and 3ʹ untranslated regions (UTR) were cloned using the 5′ and 3′ Full RACE Kit (TaKaRa, Kusatsu, Japan) according to the manufacturer’s protocol. The PCR fragments were purified by gel extraction and then were incubated with linearized pMD18-T vector and T4 ligase at 16 °C overnight. The recombinant vectors were transfected to *E. coli* DH5α (TransGen, Beijing, China), and then monoclonal *E. coli* were sequenced by Tsing Ke company. Finally, the *CiBATF3* full length cDNA was assembled using DNAMAN software.

### 4.4. Sequence Analysis

The Sequence Manipulation Suite (STS) (http://www.bio-soft.net/sms/) were used to analyze the amino acid sequence and nucleotide sequence of *CiBATF3*. The *CiBATF3* full-length cDNA sequence was aligned to the genomic sequence using BLAST to determine the intron/exon structure of the genomic sequence. The protein domain characteristics were analyzed using the Simple SMART program (http://smart.emblheidelberg.de/) and the CDD tool (http://www.ncbi.nlm.nih.gov/Structure/cdd/wrpsb.cgi). The multiple sequence alignment was performed with ClustalW2 (http://www.ebi.ac.uk/Tools/msa/clustalw2/) tool. The the amino acid sequence and nucleotide sequence from other species were collected by NCBI. Gene synteny relationship of BATF3 genes were obtained by GnomicusV88.01 [34], and then these genes used as references were compared with grass carp genomic sequence. The signal peptide was predicted using the SignaIP 4.0 Server (http://www.cbs.dtu.dk/services/SignalP/). The tertiary structure was predicted using SWISS-MODEL (http://swissmodel.expasy.org/) and displayed by Swiss-PdbViewer Version 4.1. The phylogenetic tree of CiBATF3 was constructed by MEGA v 5.0 program based on neighbor-joining (NJ) method [56]. The amino acid sequence of *CiBATF3* was comparatively analyzed with orthologous sequences by Geneious software to assess the similarity percentages.

### 4.5. Tissue Distribution of CiBATF3

To understand the expression profile of *CiBATF3* in different tissues, quantitative real-time PCR (RT-qPCR) was carried out using iQ™ SYBR Green Supermix (Bio-Rad, Hercules, CA, USA ) on a CFX96^TM^ Real Time Detection System (Bio-Rad, Hercules, CA, USA). *Ctenopharyngodon idella* β-actin (Accession No. M25013.1) was selected as reference gene. The specific primers are listed in Table 2. Finally, the Ct values for respective reaction were subjected to comparative Ct method (2^−ΔΔ*C*T^) [57] to calculate the *CiBATF3* mRNA levels. Data are presented as the *CiBATF3* expression-fold relative to that of β-actin mRNA and are presented as mean ± standard deviation (SD).

### 4.6. Responses of CiBATF3, IRF8 and IL-10 Genes to GCRV Infection In Vivo

RT-qPCR was employed for examining the responses of *CiBATF3*, IRF8 and IL-10 following GCRV infection. β-actin (Accession No. M25013.1) was selected as internal control. Relative expression levels were calculated as the ratio of gene expression in the infected grass carp at each time point (1, 3 and 5 days after GCRV infection) relative to that in the uninfected fish (Day 0). To test the effectiveness of GCRV infection, the relative copy numbers of the virus in gill and intestine were examined by RT-qPCR. The expression level of Day 1 was set as the baseline (1.0). Special primers were designed based on the S6 segments of GCRV-II (see Table 2).

### 4.7. The mRNA Expression Profiles of CiBATF3 Gene in CIK Cells Following Poly(I:C) Challenge, LPS Stimulation

To further investigate the effect on *CiBATF3* in vitro post PAMPs challenges, the expression patterns of RT-qPCR were employed. For LPS stimulation, cells were exposed to 10 μg mL^−1^ LPS (*Escherichia coli* 055: B5, Sigma, USA) for 1, 2, 6, 12, 24 and 48 h. Untreated cells served as controls. For poly(I:C) challenge, cells were exposed to 20 μg mL^−1^ poly(I:C) (sigma, USA) for 0, 3, 8, 24, 36, and 48 h. The cells at 0 h served as controls. Three biological replicate cells were collected and lysed in Trizol for subsequent RT-qPCR analysis.

### 4.8. CiBATF3 Gene Silencing Using siRNA In Vitro and In Vivo

Small interfering RNA (siRNA) that specifically targets *CiBATF3* and negative control were synthesized by GenePharma (Shanghai, China). The siRNA sequence information is listed in Table 2. For transfection experiments in vitro, 5 μL *CiBATF3* specific siRNA (20 mM) or negative controls were transfected into CIK cells using Lipo6000™ Transfection Reagent (Beyotime). At 24 h post-transfection, the treated and control CIK cells were collected for expression analysis. Three biological replicates were obtained for each group.

The transfection experiments in vivo were performed as described previously [58] with some modifications. Briefly, 10 μL of *CiBATF3* specific siRNA (20 mM) or the negative control was mixed with 10 μL of Lipo6000™ Transfection Reagent and 80 μL of PBS to serve as the transfection solution. Grass carp (weight, 10 ± 2 g; length, 7 ± 3 cm) were injected intraperitoneally with 100 μL of above transfection solution. After 24 h transfection, the head kidney and middle kidney tissues from treated or control group were collected for expression analysis using RT-qPCR. There were three biological replications at each time point.

### 4.9. Dual-Luciferase Activity Assays

The ORF sequence encoding the mature peptide of *CiBATF3* was amplified by specific primers (Table 3) with *Xho* I/*EcoR* I sites, and then ligated into pCDNA3.1 (-) eukaryotic expression vector. All recombinant vectors were transformed into *E. coli* DH5a and confirmed by sequencing. Then, positive plasmids were extracted by Endo-free plasmid mini kit I.

The process of luciferase assays was performed as described previously [26]. For the transient transfection assays, HEK293 cells were transfected with 0.25 μg pCDNA3.1-CiBATF3, 0.25 μg reporter gene vectors (pNFқB-TA-luc and pAP1-TA-luc, respectively), 0.025 μg pRL-TK renilla luciferase vector and 1 μL Lipo6000™ Transfection Reagent. At 24 h post- transfection, the cells were collected and lysed. Luciferase activities were measured using the Luciferase Assay System (Promega, Wisconsin, USA). The experiments were repeated at least three times. Differences were considered significant at *p* < 0.05.

### 4.10. Subcellular Localization of CiBATF3 in CIK Cells and HEK293 Cells

Specific primer pairs containing *XhoI* and *BamHI* restriction enzyme cutting sites (Table 3) were designed to amplify the complete ORF sequence of CiBATF3. After being digested by restriction enzyme, the PCR product was cloned into pEGFP-N3 vectors producing GFP-tagged expression plasmids. Subsequently, CIK cells and HEK293T cells were seeded onto a sterile microscope cover glass and placed in a 6-well cell culture plate prior to transfection. After cells were grown to 70–90% confluence, 5 μg of pEGFP-N3-CiBATF3 vectors were transiently transfected, while empty pEGFP-N3 vector was used as the control. After transfection for 24 h, CIK cells and HEK293T cells were washed three times with PBS and fixed with 4% (*v*/*v*) paraformaldehyde for 15 min at room temperature. Then, CIK cells and HEK293T cells were dyed with Hoechst 33342 (Beyotime, China) and observed under the UltraVIEW VOX confocal system (PerkinElmer, Fremont, CA, USA) and a 63× oil immersion objective lens.

To evaluate the influence of PAMPs stimulation on subcellular localization of CiBATF3 proteins, HEK293T cells were transfected with CiBATF3-EGFP fusion plasmid or control plasmid (pEGFP-N3) and seeded into six-well plates. Approximately 24 h later, the cells were treated with 50 mg/mL poly(I:C) and 20 mg/mL LPS, respectively. After PAMPs stimulation, the HEK293T cells were observed using confocal system.

### 4.11. Verification of the Protein Interaction between CiBATF3 and IL-10

Interleukin-10 (IL-10) is known as an anti-inflammatory cytokine, which plays critical inhibitory roles in broad spectrum of immune responses [59]. It was predicted by online InBio Map™ version 2016_09_12 (https://www.intomics.com/inbio/map.html) that BATF3 interacts with IL-10. The bimolecular fluorescence complementation (BiFC) system is based on the reconstitution of two non-fluorescent fragments of a fluorescent protein, which could be used to determine whether two proteins are interactive [60,61]. In the present study, the mNeptune-based BiFC system was introduced to visualize whether IL-10 interacts with CiBATF3 in CIK cells. Briefly, the ORF sequences of CiBATF3 and IL-10 (amplified by primers in Table 3) were cloned from grass carp and inserted into the pMN155 and pMC156 plasmids, respectively. The final plasmids were named pBATF3-MN155 and pMC156- IL-10, which contained the *N*-terminal of mNeptune (mNeptune aa 1–155, MN155) and *C*-terminal of mNeptune (mNeptune aa 156C-terminal, MC156), respectively. Then, plasmids pBATF3-MN155 and pMC156- IL-10 were transfected into CIK cells alone or together. At 24 h post-transfection, cells were observed as described above.

### 4.12. Statistical Analysis

All the data were analyzed using the SPSS software (version 22.0 USA) and assessed by one-way analysis of variance (ANOVA). All the experiments were repeated at least three times. The significance level was set at *p* < 0.05 and the highest significance level was set at *p* < 0.01.

## Figures and Tables

**Figure 1 ijms-20-01687-f001:**
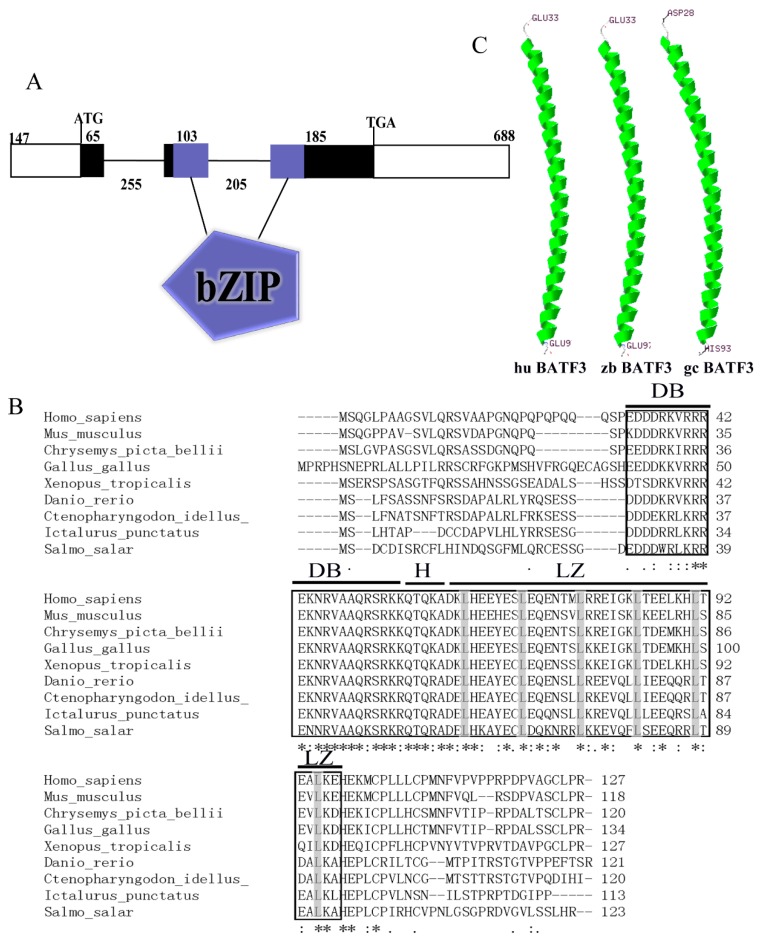
(**A**) Prediction of domain and gene structure of CiBATF3 by the online website SMART. The lines represent introns; boxes represent exons; and numbers represent the base pair length of each region; and bZIP represent basic leucine zipper domain. (**B**) Amino acid sequence alignment of CiBATF3 with other species BATF3. The amino acid sequences were aligned using the ClustalW 2.1 program. The core domains were framed with square, including leucine zipper (LZ) regions, hinge (H) and DNA binding domain (DB). Conserved leucine residues are highlighted in gray. Below the alignment, identity (*), strong similarity (:) and weak similarity (.) are represented. (**C**) The spatial structure of BATF3 protein from human, zebrafish and grass carp were predicted by SWISS-MODEL program. Green, α- helices.

**Figure 2 ijms-20-01687-f002:**
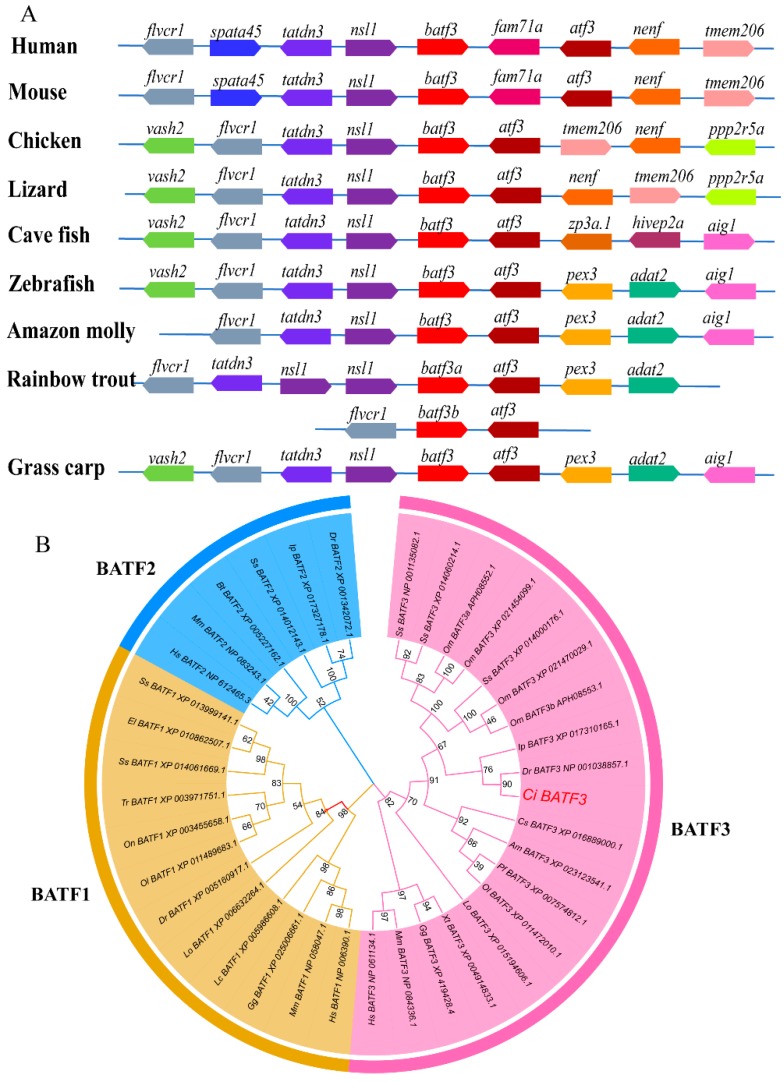
(**A**) Syntenic analysis of BATF3 gene in *C. idella* with different species. Gene synteny relationship of BATF3 genes were obtained by GnomicusV88.01 [34], and then these genes used as references were compared with grass carp genomic sequence. The syntenic analysis of BATF3 genes was performed with neighboring genes from other species. The arrows indicate the direction of transcription. Homologous genes are displayed in the same color. (**B**) The phylogenetic tree of BATF family genes. The phylogenetic tree of BATF3 was constructed by MEGA v 5.0 program based on neighbor-joining (NJ) method. Abbreviations of the species presented in the figure are as follows: *Mus musculus* (Mm), *Oryzias latipes* (Ol), *Bos taurus* (Bt), *Takifugu rubripes* (Tr), *Gallus gallus* (Gg), *Xenopus tropicalis* (Xt), *Danio rerio* (Dr), *Oncorhynchus mykiss* (Om), *Homo sapiens* (Hs), *Poecilia formosa* (Pf), *Salmo salar* (Ss), *Latimeria chalumnae* (Lc), *Lepisosteus oculatus* (Lo), *Oreochromis niloticus* (On), *Esox lucius* (El), *Ictalurus punctatus* (Ip), *Cynoglossus semilaevis* (Cs). The accession numbers are given after the species names in the tree.

**Figure 3 ijms-20-01687-f003:**
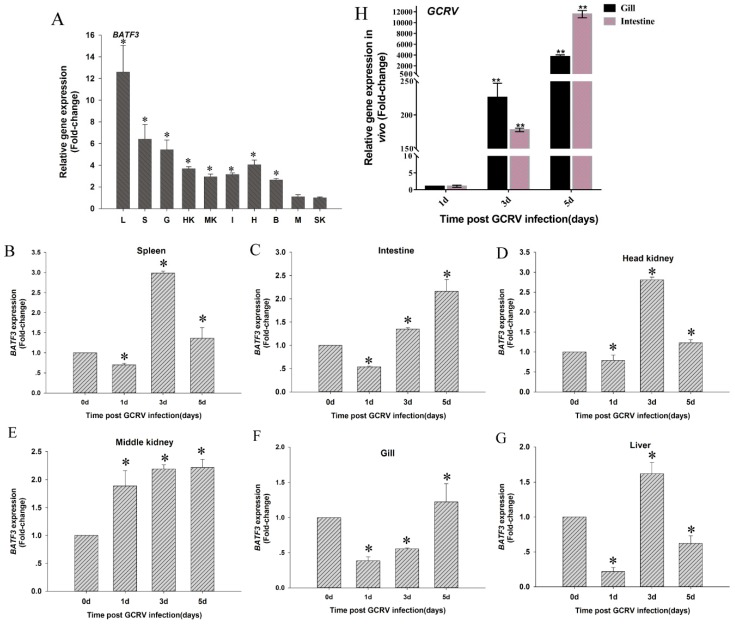
(**A**) Tissue distribution of *CiBATF3*; (**B**–**G**) the expression patterns of *CiBATF3* in immune-related tissues after GCRV challenge; and (**H**) the expression of the S6 gene in gill and intestine. The expression level of the control group was set as the baseline (1.0). The results are presented as the mean ± SD (*n* = 5). The abbreviations of 10 examined tissues. L, liver; S, spleen; G, gill; HK, head kidney; B, brain; MK, middle kidney; H, heart; M, muscle; SK, skin; I, intestine. * = *p* ≤ 0.05, ** = *p* ≤ 0.01.

**Figure 4 ijms-20-01687-f004:**
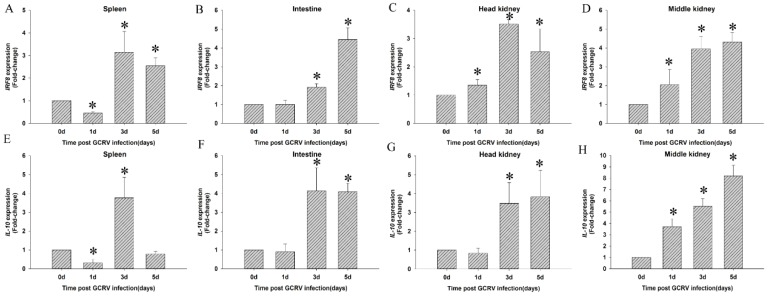
The expression levels of IRF8 (**A**–**D**) and IL-10 (**E**–**H**) in four immune tissues (spleen, intestine, head kidney and middle kidney) after GCRV infection. The results are presented as the mean ± SD (*n* = 5). The expression level of the control group was set as the baseline (1.0) and were normalized to the expression in the untreated groups (Day 0). * = *p* ≤ 0.05.

**Figure 5 ijms-20-01687-f005:**
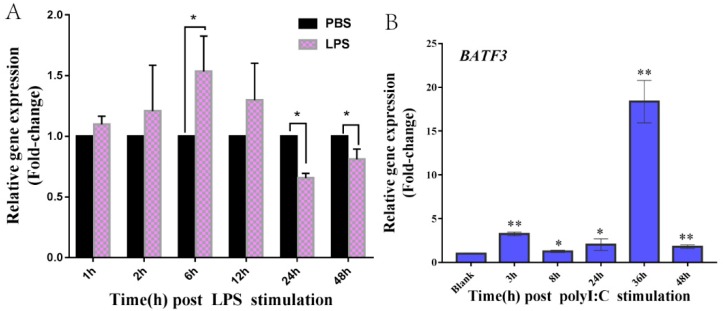
The expression patterns of *CiBATF3* in CIK cells after LPS (**A**) or poly(I:C) (**B**) challenge. CIK cells were stimulated with poly(I:C) (20 μg mL^−1^) and LPS (10 μg mL^−1^), respectively. Untreated cells served as controls. The expression level of the control group was set as the baseline (1.0) and the results are presented as the mean ± SD (*n* = 3). * = *p* ≤ 0.05, ** = *p* ≤ 0.01.

**Figure 6 ijms-20-01687-f006:**
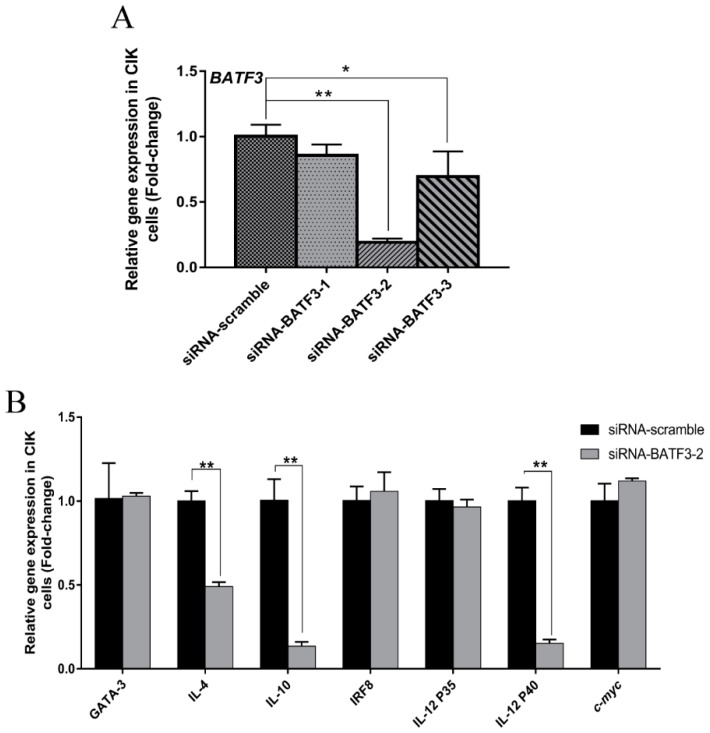
Gene expression profiles in *CiBATF3* deficient CIK cells generated by siRNA. (**A**) Three siRNAs targeted to different sites of *CiBATF3* were used to infected CIK cells, and siRNA-BATF3-2 has the highest efficiency in *CiBATF3* gene silencing. (**B**) The expression of *CiBATF3*, GATA3, IL-4, IL-10 IRF8, IL-12 p35, IL-12 p40 and *c-myc* at 24 h were assayed by RT-qPCR after infected with siRNA-BATF3-2. The expression level of the control group was set as the baseline (1.0) and the results are presented as the mean ± SD (*n* = 3). siRNA-scramble: control siRNA infection; siRNA-BATF3: *CiBATF3* siRNA infection. * = *p* ≤ 0.05, ** = *p* ≤ 0.01.

**Figure 7 ijms-20-01687-f007:**
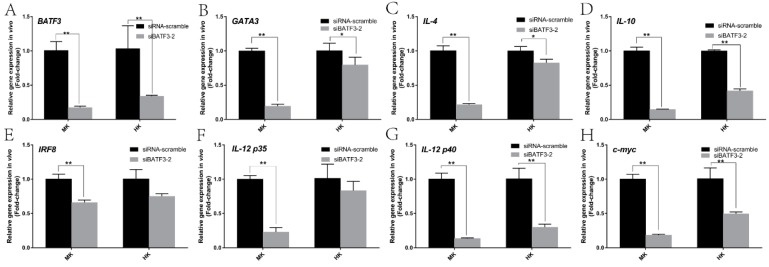
Gene expression profiling in head kidney (HK) and middle kidney (MK) at 24 h after injected siRNA-*CiBATF3* in vivo. The expression of *CiBATF3* (**A**), GATA3 (**B**), IL-4 (**C**), IL-10 (**D**), IRF8 (**E**), IL-12 p35 (**F**), IL-12 p40 (**G**) and *c-myc* (**H**) at 24 h were assayed by RT-qPCR. The expression level of the control group was set as the baseline (1.0) and the results are presented as the mean ± SD (*n* = 3). siRNA-scramble: control siRNA infection; siBATF3: *CiBATF3* siRNA infection. * = *p* ≤ 0.05, ** = *p* ≤ 0.01.

**Figure 8 ijms-20-01687-f008:**
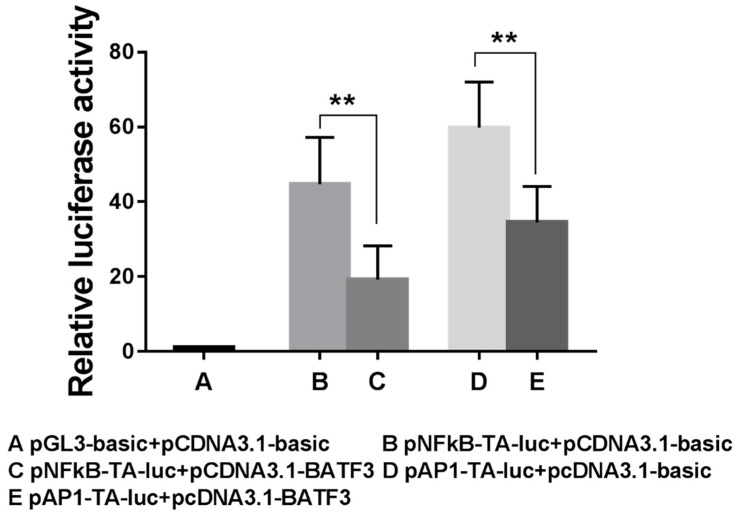
Impact of CiBATF3 on AP-1 and NF-κB reporter gene. HEK293 cells were transfected with 0.25 μg pCDNA3.1-CiBATF3, 0.25 μg reporter gene vectors (pNFқB-TA-luc and pAP1-TA-luc, respectively), 0.025 μg pRL-TK renilla luciferase vector and 1 μL Lipo6000™ Transfection Reagent. At 24 h post-transfection, the cells were collected and lysed. Luciferase activities were measured using the Luciferase Assay System (Promega). The results are presented as the mean ± SD (*n* = 3). Differences were considered significant at *p* < 0.01.

**Figure 9 ijms-20-01687-f009:**
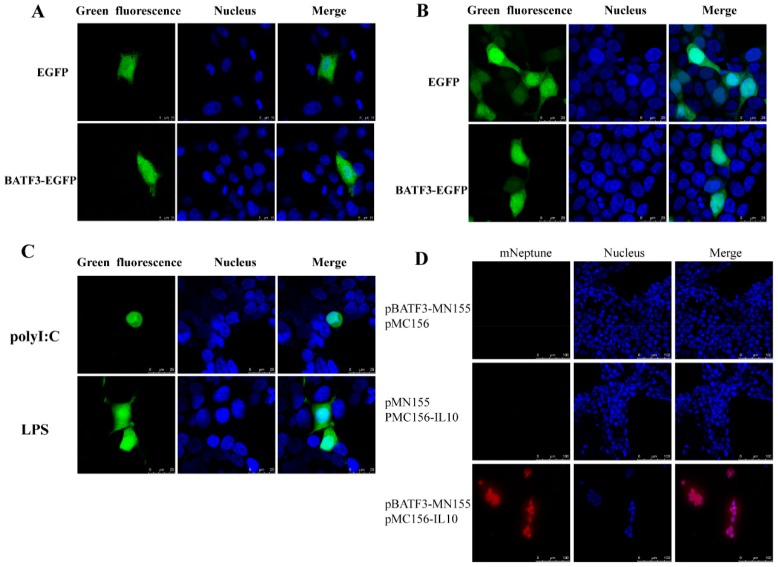
Subcellular localization of CiBATF3 in: CIK cells (**A**); and HEK293 cells (**B**). (**C**) CiBATF3-GFP nuclear translocation induced by PAMPs stimulation. Green fluorescence shows the distribution of EGFP or EGFP-tagged proteins and blue fluorescence shows the nucleus that was stained by Hoechst 33342 under a 60× oil immersion objective lens (scale bars, 10 μm and 25 μm). (**D**) Imaging of the protein–protein interaction by using far-red mNeptune-based BiFC in HEK293 cells. The fluorescence of the mNeptune channel was red. The images were acquired using fluorescence microscopy and a 40× oil immersion objective lens (scale bar, 100 μm).

**Table 1 ijms-20-01687-t001:** Amino acid sequence similarity of *CiBATF3* with other species by Geneious.

	The Species Name and Accession Numbers	Percent of Identity																	
**Percent of identity**		1	2	3	4	5	6	7	8	9	10	11	12	13	14	15	16	17	18
	1. *CiBATF3*		85	69.8	50.8	50.8	51.6	53.6	50	50	44.5	48.7	41.7	44.1	43.3	36.9	40.3	39.7	46.9
	2. DrBATF3_NP_001038857.1	85		67.2	52	52.8	52	53.1	51.2	51.2	44.5	47.9	40.2	42.4	44.1	36.9	41.1	40.5	45.3
	3. IpBATF3_XP_017310165.1	69.8	67.2		49.2	50	53.4	54.6	51.6	51.6	50	48.7	45.1	43	41.5	37.3	41.6	42.1	47.2
	4. OmBATF3a_APH08552.1	50.8	52	49.2		98.4	89.4	80.5	69	69	48.8	53.7	41.7	41.7	33.9	31.5	32.6	31	42.2
	5. OmBATF3_XP_021454099.1	50.8	52.8	50	98.4		89.4	80.5	69	69	47.9	52.9	40.9	40	34.6	32.3	33.3	31.7	42.2
	6. SsBATF3_XP_014060214.1	51.6	52	53.4	89.4	89.4		88.5	73.6	73.6	49.6	54.5	44.1	42.5	37	35.4	36.4	35.7	46.9
	7. SsBATF3_NP_001135082.1	53.6	53.1	54.6	80.5	80.5	88.5		69.7	68.9	51.4	52.3	44.4	43.1	36.8	37.5	37.8	37.4	47.5
	8. OmBATF3b_APH08553.1	50	51.2	51.6	69	69	73.6	69.7		96.9	50	56.3	46.6	41.6	39.8	35.9	37.7	36.2	48.1
	9. SsBATF3_XP_014000176.1	50	51.2	51.6	69	69	73.6	68.9	96.9		49.2	54.8	45	40	39.1	36.6	36.2	35.4	48.1
	10. PfBATF3_XP_007574812.1	44.5	44.5	50	48.8	47.9	49.6	51.4	50	49.2		70.6	61.3	53	38.1	34.4	37.5	38.7	38.9
	11. AoBATF3_XP_023123541.1	48.7	47.9	48.7	53.7	52.9	54.5	52.3	56.3	54.8	70.6		63	54.2	34.9	33.3	38.3	39.2	40.2
	12. OlBATF3_XP_011472010.1	41.7	40.2	45.1	41.7	40.9	44.1	44.4	46.6	45	61.3	63		52	35.1	32.8	40.6	38	42
	13. CsBATF3_XP_016889000.1	44.1	42.4	43	41.7	40	42.5	43.1	41.6	40	53	54.2	52		35.2	30.5	37.6	35.5	41.8
	14. XtBATF3_XP_004914833.1	43.3	44.1	41.5	33.9	34.6	37	36.8	39.8	39.1	38.1	34.9	35.1	35.2		61.5	61.2	56.6	53.1
	15. GgBATF3_XP_419428.4	36.9	36.9	37.3	31.5	32.3	35.4	37.5	35.9	36.6	34.4	33.3	32.8	30.5	61.5		61.5	57.7	54.1
	16. HsBATF3_NP_061134.1	40.3	41.1	41.6	32.6	33.3	36.4	37.8	37.7	36.2	37.5	38.3	40.6	37.6	61.2	61.5		80.3	57.5
	17. MmBATF3_NP_084336.1	39.7	40.5	42.1	31	31.7	35.7	37.4	36.2	35.4	38.7	39.2	38	35.5	56.6	57.7	80.3		55.2
	18. LoBATF3_XP_015194606.1	46.9	45.3	47.2	42.2	42.2	46.9	47.5	48.1	48.1	38.9	40.2	42	41.8	53.1	54.1	57.5	55.2	

Abbreviations of the species presented in the figure are as follows: *Ctenopharyngodon idella* (Ci), *Amphiprion ocellaris* (Ao), *Mus musculus* (Mm), *Oryzias latipes* (Ol), *Bos taurus* (Bt), *Gallus gallus* (Gg), *Xenopus tropicalis* (Xt), *Danio rerio* (Dr), *Oncorhynchus mykiss* (Om), *Homo sapiens* (Hs), *Poecilia formosa* (Pf), *Salmo salar* (Ss), *Lepisosteus oculatus* (Lo), *Ictalurus punctatus* (Ip), *Cynoglossus semilaevis* (Cs). The accession numbers are given after the species names in the table.

**Table 2 ijms-20-01687-t002:** Primers used for cloning and RT-qPCR and the siRNA sequence information.

Primers	Sequences (5′—3′)	Purpose
BATF3-5′Rout	GCACTGTGCCTGTGGACCTT	5′ RACE
BATF3-5′Rin	ACGCCTCGTGCAACTCGTCA	
BATF3-3′Rout	AGTGATGCTCCAGCTTTACGGT	3′ RACE
BATF3-3′Rin	ACCGAGTTGCTGCCCAGAGA	
BATF3-F	ATGTCACTTTTCAATGCGACAAGTAA	cDNA cloning
BATF3-R	TCAGATGTGAATGTCTTGTGGCACTGT	
qBATF3-F	AGTGATGCTCCAGCTTTACGGT	RT-qPCR
qBATF3-R	ACGCCTCGTGCAACTCGTCA	
qIL10-F	TATTAAACGAGAACGTGCAACAGAA	
qIL10-R	TCCCGCTTGAGATCTTGAAATATACT	
qIRF8-F	CAGAGGAGGAACAGAAGTTGGGTAA	
qIRF8-R	ACGCTTCAGGATGCCCATGTA	
q β-actin-F	TCGGTATGGGACAGAAGGAC	
q β-actin-R	GACCAGAGGCATACAGGGAC	
qIL4-F	CTCAGGTGAAGCCCTTTGCC	
qIL4-R	ACTGGATGTTCCTCTGAAGCTGTAA	
qIL-12 p35-F	AGGCTCGGATGATTCCTTACA	
qIL-12 p35-R	TCACACTGGGCTGGTAGGAG	
qIL-12 p40-F	GGAGAAGTCTACGAAGGGCAA	
qIL-12 p40-F	GTGTGTGGTTTAGGTAGGAGCC	
q c-myc-F	GAGCGAAGACATTTGGAAGA	
q c-myc-R	TGATGAAGGACTGGGAGTAG	
qGATA-3-F	TACGAGGAGGACAAAGAGT	
qGATA-3-R	GTAAGTGGCGATGGGATGGT	
qS6-F	AGCGCAGCAGGCAATTACTATCT	GCRV RT-qPCR
qS6-R	ATCTGCTGGTAATGCGGAACG	
Negative control	UUCUCCGAACGUGUCACGUTT	siRNA
siRNA-BATF3-1	GGCGUAUGAGUGUCUGGAATT	
siRNA-BATF3-2	GAGGAACAGCAACGCUUAATT	
siRNA-BATF3-3	GCACAGUGCCACAAGACAUTT	

**Table 3 ijms-20-01687-t003:** Primers used in plasmid construction. The different restriction sites are shown in lowercase letters and the names are represented above the respective nucleotides.

Primers	Sequences (5′–3′)	Purpose
L-BATF3-F	*XhoI*CGctcgagGTATGTCACTTTTCAATGCGACAAGTAA	Subcellular localization
L-BATF3-R	*BamHI*CGggatccGATGTGAATGTCTTGTGGCACTGT	
BF-BATF3-F	*XhoI*CCCctcgagCTATGTCACTTTTCAATGCGACAAGTAA	BiFC Analysis
BF-BATF3-R	*EcoRI*CAAgaattcGA*ACTCCCGCCACCTCCACTCCCG**CCACCTCC*GATGTGAATGTCTTGTGGCACTGT	
BF-IL10-F	*EcoRI*CGgaattcTG*GGAGGTGGCGGGAGTGGAGGTGGCGG**GAGT*ATGATTTTCTCTAGAGTCATCTTTTCTGC	
BF-IL10-R	*KpnI*CGCggtaccGTGCTTTTCTCTCTTTGATGCCAG

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
