# Peer review of "Investigating the Role of BATF3 in Grass Carp (Ctenopharyngodon idella) Immune Modulation: A Fundamental Functional Analysis"

_ijms, 2019, doi:10.3390/ijms20071687_

Reviewer 1 Report

The authors describe how the transcription factor BATF3, an essential cDC1 differentiation, is found in abundance in immune related organs of carp. The factor is found throughout cells and translocates from the nucleus upon DC activation via TLR agonist stimulation. Some main transcription factors and cytokines that are involved BATF3 responsiveness/cell sensing are identified. Overall I would like to congratulate the authors for their study which logically describes the role and distribution of BATF3 in carp, especially considering the relatively low amount of information about fish innate immune cells compared to other experimental animals. Specific comments are found below:

 Line 63: please add more citations to the ‘many studies’ 

 Table 1: in the footnotes of the table add the full names of the species that have been abbreviated to two letters in the table. Currently the reader must use Figure 1B to know what species the BATF3 comes from.   

 Line 107 to 110, some of the text appears redundant, please verify.

 Figure 3. 

If showing individual fish and not individual experimental repeats, the errors bars should be shown as S.D. rather than S.E.M.

The paragraphs in the discussion are much too long. Please divide the large paragraphs into several smaller paragraphs when possible (one general idea per paragraph).

 When possible, each results section paragraph should have a brief sentence at the beginning and end to help link it to the paragraphs before and after. This will help the flow of the paper, and help people who are not experts in your field and transcription factors to understand the logic of your study plan. In other words, there need to be more details about why you chose to do certain types of experiments and only look at certain factors or readouts – which can be used to introduce each results section. Conversely, at the end of each results section, one or more concluding generalized statements would also be very helpful to assist people in understanding the more technically-dense paragraphs. For example, lines 314 to 318 could probably be moved to the results, and save the discussion section to place your results in context with other studies, or to propose new hypotheses, limitations of the study, describe new mechanistic or functional insights into molecular pathways or tissue distribution of cell subsets/transcription factors etc.

 Line 303 and 313, it would be more precise to quote the TLRs that recognize the PAMPs you mention, rather than use a very generalized statement using the words ‘viral and bacterial’ (because you did not test all of the known viral and bacterial PAMPs). Also, you might want to cite studies that characterize the main TLRs expressed by cDC1, and state whether your PAMP responsiveness findings support or don’t support these previous findings.

Author Response

Response to Reviewer 1 Comments

Point 1: Line 63: please add more citations to the ‘many studies’ 

Response 1: We have added more citations to the ‘many studies’ in Line 84.

Point 2: Table 1: in the footnotes of the table add the full names of the species that have been abbreviated to two letters in the table. Currently the reader must use Figure 1B to know what species the BATF3 comes from.

Response 2: What you mean is that Table 3 should add the full names of the species that have been abbreviated to two letters in the table. Table 1 is shown the primers sequence information. We have added the full name of the species to the footnotes of the Table 3.

Point 3: Line 107 to 110, some of the text appears redundant, please verify.

Response 3:  We have modified these sentences in Lines 144 to 148.

Point 4: Figure 3. If showing individual fish and not individual experimental repeats, the errors bars should be shown as S.D. rather than S.E.M.

Response 4: We have modified these mistakes in Figures 3 to 7.

Point 5: The paragraphs in the discussion are much too long. Please divide the large paragraphs into several smaller paragraphs when possible (one general idea per paragraph).

    When possible, each results section paragraph should have a brief sentence at the beginning and end to help link it to the paragraphs before and after. This will help the flow of the paper, and help people who are not experts in your field and transcription factors to understand the logic of your study plan. In other words, there need to be more details about why you chose to do certain types of experiments and only look at certain factors or readouts – which can be used to introduce each results section. Conversely, at the end of each results section, one or more concluding generalized statements would also be very helpful to assist people in understanding the more technically-dense paragraphs. For example, lines 314 to 318 could probably be moved to the results, and save the discussion section to place your results in context with other studies, or to propose new hypotheses, limitations of the study, describe new mechanistic or functional insights into molecular pathways or tissue distribution of cell subsets/transcription factors etc.

Response 5: We have revised the long paragraphs in the discussion according to the comments of Reviewer 1, please see “discussion”.

Point 6:  Line 303 and 313, it would be more precise to quote the TLRs that recognize the PAMPs you mention, rather than use a very generalized statement using the words ‘viral and bacterial’ (because you did not test all of the known viral and bacterial PAMPs). Also, you might want to cite studies that characterize the main TLRs expressed by cDC1, and state whether your PAMP responsiveness findings support or don’t support these previous findings.

Response 6: We have rewritten the paragraph again, please see Lines 343 to 355.

Reviewer 2 Report

Zhu and colleagues investigated the role of BATF3 in the immune system of grass carps. In general, they performed a well designed study employing sophisticated methods, which is interesting for specialised immunologists. However, the manuscript is written quite carelessly and contains many typing errors (e.g. „BAT3“ in line 280, „Flod change“ on the Y-axis in Figures 3 and 4; „BTAT3 gene“ in Lin 424, „CiBTAT3“ in line 426) and even redundant sentences (e.g. lines 86-89, lines 105-109). Moreover, the englisch is not adequate making it difficult to understand the meaning of some sentences. Abbreviations are not always introduced properly (e.g. "CIK cells" in the abstract; „“BLRZ“ in Figure 1). The authors use the abbreviations "BATF3“ and „CiBTAF3“ in a random way.

Additional points:

1. The authors should revise their introduction to better explain known functions of IRF8 and the difference between CD8alpha+ classical dendritic cells (DCs), CD03+ DCs, migratory DCs, and other antigen presenting cells.

2. Line 65: What is exactly meant by „may be compensated“.

3. Line 76: Do the authors mean „transcriptional regulation of BATF3“ or "transcriptional regulation by BATF3“?

4. Lines 140-143: „heart“ is missing in the description. Also missing in lines 363-365.

5. Line 166: There are no "**“ in Figure 4.

6. Figure 5B: Is there a statistically significant difference at 24h?

7. Figure 6B: The authors should better explain why they investigated „GATA3, IL-4, IL-10, IRF8, IL-12 p35, IL-12 p40, c-myc“.

8. Line 200: Not ALL genes were down-regulated in middle kidney and head kidney (e.g. IRF8, IL-12 p35).

9. Line 223: p values between „B“ and „C“ as well as „D“ and „E“ are below 0.01 and not only 0.05.

10. Line 264: According to Figure 3A the correct order should be „liver, spleen, gills, heart, and head kidney“.

11. Line 327-328: The authors describe an interaction between the BATF3 protein and the IL-10 protein in the cytoplasm of CIK cells. What do they mean exactly ? Is there anything known about interactions between AP-1 transcription factors and cytokine proteins? To the best of my knowledge only canonical pathways of AP-1 transcription factors binding to the promoter (DNA) of cytokines have been described so far.

12. Figure 9C: Which cell type is shown in this Figure (CIK cells, HEK293 cells, something else)? Please add the respective description to „Materials and methods“ (lines 461-476).

13. lines 370-372: Why did the authors keep the infected fish without food for 6 days?

14. lines 385-386: How were the amplified fragments inserted into the pMD18-T vector? The authors should extend the description of their cloning procedures.

15. lines 420-421: Figure 3H also shows quantification of GCRV RNA in the intestines. Please add „intestines“to the respective parts of „Materials and methods“.

16. Table 2: Please add the names of the different restriction sites (underlined in the different primers) above the respective nucleotides (XhoI, BamHI, …).

Author Response

Response to Reviewer 2 Comments

Point 1: the manuscript is written quite carelessly and contains many typing errors (e.g. „BAT3“ in line 280, „Flod change“ on the Y-axis in Figures 3 and 4; „BTAT3 gene“ in Lin 424, „CiBTAT3“ in line 426) and even redundant sentences (e.g. lines 86-89, lines 105-109). Moreover, the englisch is not adequate making it difficult to understand the meaning of some sentences. Abbreviations are not always introduced properly (e.g. "CIK cells" in the abstract; „“BLRZ“ in Figure 1). The authors use the abbreviations "BATF3“ and „CiBTAF3“ in a random way.

Response 1: “Flod change” on the Y-axis in Figures 3 and 4 has been modified to “Fold change”; Redundant sentences have been deleted; Abbreviations introduced inproperly have been modified; The BATF3 gene name in grass carp is abbreviated as CiBATF3 in this article; Some sentences in the article have been modified.

 Point 2: The authors should revise their introduction to better explain known functions of IRF8 and the difference between CD8alpha+ classical dendritic cells (DCs), CD03+ DCs, migratory DCs, and other antigen presenting cells.

Response 2: We have modified the introduction according to the comments of Reviewer 2, please see Line 62 to 77 in yellow background.

 Point 3: Line 65: What is exactly meant by „may be compensated“.

Response 3: It means that BATF1 and BATF2 could replace BATF3 for cDC development.

 Point 4: Line 76: Do the authors mean „transcriptional regulation of BATF3“ or "transcriptional regulation by BATF3“?

Response 4: Sorry, our expression is inappropriate. Its mean is “transcriptional regulation by BATF3”

Point 5: Lines 140-143: „heart“ is missing in the description. Also missing in lines 363-365.

Response 5: We have modified these mistakes in Lines 140-143 and lines 363-365.

 Point 6: Line 166: There are no "**“ in Figure 4.

Response 6: The "**“ have been deleted from the footnote in Figure 4.

 Point 7: Figure 5B: Is there a statistically significant difference at 24h?

Response 7: There is a statistically significant difference at 24h (P ≤ 0.05) and Figure 5B have been revised.

 Point 8: Figure 6B: The authors should better explain why they investigated „GATA3, IL-4, IL-10, IRF8, IL-12 p35, IL-12 p40, c-myc“.

Response 8: We have added the reason for the investigation in lines 292 to 293, lines 414 to 424.

 Point 9: Line 200: Not ALL genes were down-regulated in middle kidney and head kidney (e.g. IRF8, IL-12 p35).

Response 9:  Most of its downstream genes were significantly down-regulated in middle kidney and head kidney except that IRF8 and IL-12 p35 did not change significantly in head kidney at this condition (Fig. 7B-H), respectively.

 Point 10: Line 223: p values between „B“ and „C“ as well as „D“ and „E“ are below 0.01 and not only 0.05.

Response 10: We have revised the mistake Line 223.

 Point 11: Line 264: According to Figure 3A the correct order should be „liver, spleen, gills, heart, and head kidney“.

Response 11: The correct order has been modified in the article in line 188.

 Point 12: Line 327-328: The authors describe an interaction between the BATF3 protein and the IL-10 protein in the cytoplasm of CIK cells. What do they mean exactly? Is there anything known about interactions between AP-1 transcription factors and cytokine proteins? To the best of my knowledge only canonical pathways of AP-1 transcription factors binding to the promoter (DNA) of cytokines have been described so far.

Response 12: We should discuss the AP-1 reporter gene in a separate paragraph, please see Lines 427 to 435.

 Point 13: Figure 9C: Which cell type is shown in this Figure (CIK cells, HEK293 cells, something else)? Please add the respective description to „Materials and methods“ (lines 461-476).

Response 13: The cell type is HEK293T cells shown in this Figure 9C. The description has been added in Line 674.

 Point 14: lines 370-372: Why did the authors keep the infected fish without food for 6 days?

Response 14: Sorry, our meaning in the article is not clearly stated. The grass carp were fed as above with commercial feed, however, the fish did not eat the feed after infection. This sentence has been revised, please see in Line 548.

 Point 15: lines 385-386: How were the amplified fragments inserted into the pMD18-T vector? The authors should extend the description of their cloning procedures.

Response 15: The description of their cloning procedures has been extended in lines 565 to 568.

 Point 16: lines 420-421: Figure 3H also shows quantification of GCRV RNA in the intestines. Please add „intestines“to the respective parts of „Materials and methods“.

Response 16: We have added “intestines” to the respective parts of “Materials and methods”.

 Point 17: Table 2: Please add the names of the different restriction sites (underlined in the different primers) above the respective nucleotides (XhoI, BamHI, …).

Response 17: The names of the different restriction sites have been added according to the comments of Reviewer 2, please see Table 2.